# Controlling Hair Loss by Regulating Apoptosis in Hair Follicles: A Comprehensive Overview

**DOI:** 10.3390/biom14010020

**Published:** 2023-12-22

**Authors:** Wuji Wang, Honglan Wang, Yunluan Long, Zheng Li, Jingjie Li

**Affiliations:** 1Key Laboratory of Basic Pharmacology of Ministry of Education and Joint International Research Laboratory of Ethnomedicine of Ministry of Education, Zunyi Medical University, Zunyi 563006, China; wangwuji@zmu.edu.cn (W.W.); wanghonglan@zmu.edu.cn (H.W.); longyunluan@zmu.edu.cn (Y.L.); lizheng@zmu.edu.cn (Z.L.); 2Key Laboratory of Basic Pharmacology of Guizhou Province, Zunyi Medical University, Zunyi 563006, China; 3Department of Pharmacology, School of Pharmacy, Zunyi Medical University, Zunyi 563006, China

**Keywords:** apoptosis, hair follicles, molecular mechanisms, hair loss, medicine treatment

## Abstract

Apoptosis is a physiological process that occurs in all cell types of the human body, and it profoundly changes the fate of hair by affecting hair follicle cells. This review outlines the cellular changes, intrinsic biochemical characteristics, and mechanisms underlying apoptosis and summarizes the hair follicle life cycle, including development, cycle stages, and corresponding cellular changes. Finally, the relationship between apoptosis and the hair cycle is discussed and the significance of apoptosis in hair loss conditions and drug treatments is highlighted. Apoptosis induces cellular changes and exhibits distinctive properties through intricate signaling pathways. Hair follicles undergo cyclic periods of growth, regression, and dormancy. Apoptosis is closely correlated with the regression phase by triggering hair follicle cell death and shedding. Regulation of apoptosis in hair follicles plays an essential role in hair loss due to maladies and drug treatments. Mitigating apoptosis can enhance hair growth and minimize hair loss. A comprehensive understanding of the correlation between apoptosis and the hair cycle can facilitate the development of novel treatments to prevent hair loss and stimulate hair regeneration.

## 1. Introduction

Apoptosis refers to the programmed cell death that influences various biological processes. During early development, apoptosis facilitates the removal of excess cells to ensure normal development, such as the gradual disappearance of the tadpole tail and the development of human fingers from a duck-like webbed foot [1]. After maturation, apoptosis remains an important part of metabolism and is responsible for the removal of damaged and dysfunctional cells or organelles. Thus, the cells in an organism are in a state of dynamic equilibrium that maintains homeostasis and normal function.

The regulation of apoptosis primarily involves the mitochondrial and death receptor pathways; however, new evidence suggests the involvement of the endoplasmic reticulum (ER) in regulating apoptosis [2]. Exposure to external stimuli can trigger endoplasmic reticulum stress (ERS) potentially caused by the accumulation of unfolded or misfolded proteins in the ER lumen [3]. ERS-induced apoptosis is primarily mediated by four major pathways, namely *PERK*, *IRE1α* [4], *ATF6*, and Ca^2+^, which involve oxidative stress, the activation of *AKT* and *JNK*, and the direct activation of *caspase-12* [5], ultimately inducing apoptosis.

The hair follicle is a unique mammalian skin structure that plays an indispensable role in skin function and regeneration as one of the important skin appendages. The hair follicle is a microscopic organ with an extremely high capacity for self-renewal and cycling through multiple hair cycles. The hair anatomy comprises, from top to bottom, the infundibulum, isthmus, bulge region, and hair bulb, and if described simply from the inside–out, the hair shaft, inner root sheath, outer root sheath, and connective tissue sheath are included in the dermal sheath (DS) [6].

Hair loss is prevalent worldwide, with chemotherapy-induced alopecia (CIA) being the most common. Although the mechanisms underlying hair loss are not yet entirely understood, apoptosis has been implicated in the pathogenesis of hair loss, particularly in CIA. However, the association between apoptosis and other forms of hair loss, such as alopecia areata (AA), lichen planopilaris (LPP), frontal fibrosing alopecia (FFA), and androgenetic alopecia (AGA), remains unclear.

This review provides an overview of apoptosis in hair loss and its potential role in different types of alopecia, focusing on the impact of apoptosis on various cells in the hair follicle, shedding light on the complex interplay between apoptosis and hair loss (Figure 1). Furthermore, recent advances in therapies that target apoptosis to treat hair loss are discussed, offering new insights and therapeutic opportunities for combating this prevalent condition.

## 2. Morphological Changes during Apoptosis

Since the late 1800s, scientists have observed apoptosis during the development of fish neurons, a phenomenon officially named apoptosis by pathologist Kerr at the University of Aberdeen, Scotland, in 1972. Since then, researchers have been exploring the regulatory mechanisms associated with apoptosis [7]. It was not until 1999 that the mechanisms underlying apoptosis in mammalian cells became much clearer using the model organism Caenorhabditis elegans (*C. elegans*) [8]. Apoptosis is a unique and important form of “programmed” cell death. Other forms of programmed death include pyroptosis, autophagy, ferroptosis, and necrosis [9].

In the early stages of apoptosis, chromatin condensation and DNA fragmentation are accompanied by a reduction in cell size, endoplasmic reticulum expansion, and ribosome aggregation. Subsequently, cells undergo fragmentation and form sealed vesicle structures called apoptotic bodies, which are eventually taken up by other cells such as macrophages or degraded via the lysosomal pathway [10,11]. During this process, there are two morphological changes in apoptotic cells: the condensation of chromatin in the nucleus and DNA degradation, followed by the formation of apoptotic bodies. 

Furthermore, two characteristics of biochemical modifications, namely cascade activation of caspase family proteins and externalization of phosphatidylserine (PS), are also observed.

In apoptotic cells, chromatin condensation and DNA degradation are the primary nuclear changes. During the initial stages of apoptosis, chromatin condenses and aggregates at the periphery of the nuclear envelope, followed by DNA fragmentation, which culminates in the rupture of the nuclear envelope. The fragmented nuclei are then sequestered within apoptotic bodies, which can be engulfed by juxtaposed cells or degraded over time [12]. Pyknosis is the most obvious feature of apoptosis often observed using fluorescent dyes [13,14].

Another obvious change in the apoptotic process is the formation of apoptotic bodies. The plasma membrane protrusions are referred to as “popcorn cytolysis” and “blebs” [15]. Cells contract and break down into apoptotic bodies, while apoptotic cell contents, such as DNA, fragments of the nucleus, and organelles, flow into the plasma membrane bodies [16]. Apoptotic body formation does not lead to leakage of cellular contents; therefore, no inflammatory response occurs, which distinguishes it from other forms of programmed cell death, such as necrosis and autophagy [17].

## 3. Biochemical Modifications of Apoptotic Cells

Apoptotic cells undergo several biochemical modifications such as cascade activation of the cysteine protein family, protein cross-linking, DNA breakage, and phagocyte recognition. In the early 1990s, Horvitz et al. reported that the cell death abnormal 3 (CED-3) protein, a requisite for programmed cell death in Caenorhabditis elegans, is homologous to the mammalian interleukin-1 beta-converting enzyme (ICE). This seminal finding culminated in the identification of ICE as *caspase-1*, which was the first identified member of the caspase family [18].

Members of the caspase family can be divided into proapoptotic and proinflammatory subgroups based on their known functions. The known proapoptotic caspase subgroups comprise mainly *caspases-2, -3, -6, -7, -8, -9*, and *-10*, which mediate cell death and are involved in signal transduction, whereas the proinflammatory caspases are *caspases-1, -4, -5, -11*, and *-12*, which regulate cytokine maturation during inflammation [19]. Another method for classifying caspases is based on the length of the structural domain of the leader peptide, which corresponds to their position in the apoptotic signaling cascade. This classification allows for the classification of caspase into initiator caspases (*caspases-1, -2, -4, -5, -8, -9, -10, 11, -12*) and effector caspases (*caspases-3, -6*, and *-7*) [20].

During caspase-dependent apoptosis, caspases-3 and -7 cleave poly (ADP-ribose) polymerase 1 (PARP1), which is responsible for DNA damage repair and transcriptional regulation [21]. Following cleavage, the 24-kDa PARP1 fragment irreversibly binds to DNA breaks, while the 89-kDa PARP1 fragment is translocated to the cytoplasm where it activates apoptosis-inducing factor (AIF) [22]. The generated AIF then binds to the nucleus and causes DNA fragmentation [23]. However, PARP1 activation is only an alternative marker of apoptosis initiated by PARP1’s over-reaction to DNA damage, which also triggers AIF, a mode of death called parthanatos [24].

The expression of cell surface markers is another biochemical feature of apoptosis. Dying cells release “find-me” signals that facilitate recognition by macrophage receptors, including the nucleotides sphingosine 1-phosphate, CX3CL1, and lysophosphatidylcholine [25,26,27,28,29]. Macrophages engulf whole cells rather than just recognizing apoptotic bodies [30]; microglia in the brain then clear early apoptotic cells [31], and neighboring companion cells can also assume responsibility for removing dying cells, as seen in epithelial cells, endothelial cells, and fibroblasts [32,33,34].

The most classic biochemical feature is the externalization of phosphatidylserine (PS), which is the “eat-me” signal expression of apoptotic cells, and after recognition by macrophages, phagocytosis begins [35]. In normal cells, the phospholipids in the plasma membrane are asymmetrically distributed and enriched within the inner leaflet [36]. When cells undergo apoptosis, the activation of caspases cleaves flippase, disrupting homeostasis [37]. Calcium phospholipid-binding protein Annexin V derivatives that bind extensively to PS are commonly used to recognize apoptosis [38,39]. Notably, PS externalization occurs not only in apoptotic cells but also in cells of necroptosis, necrosis, and ferroptosis [40,41].

## 4. The Two Main Pathways of Apoptosis

Apoptosis involves two major pathways, the first of which involves the release of cytochrome C and the formation of apoptotic bodies through mitochondrial outer membrane permeabilization (MOMP) [42,43], which promotes intrinsic apoptosis. The second pathway involves the activation of cell surface death receptors, eliciting the oligomerization of caspase-8 and assembly of the death-inducing signaling complex (DISC), thereby directing the cell toward extrinsic apoptosis [44]. During the process of apoptosis, many proteins and transcription factors are intricately connected and intertwined, and complete apoptosis is mutually influenced.

As illustrated in Figure 2, apoptosis of cells involves the regulation of many genes. The key to extrinsic apoptosis is the activation of death receptors, while the core of intrinsic apoptosis lies in the MOMP of mitochondria. These two pathways lead cells to death coincidentally, causing apoptotic characteristics such as DNA fragmentation, formation of apoptotic bodies, and phosphatidylserine externalization.

### 4.1. Intrinsic Mitochondrial Apoptosis Pathway

The intrinsic apoptotic pathway is elicited and propelled by diverse perturbations within the microenvironment, including the deprivation of growth factors, DNA damage, endoplasmic reticulum stress, accumulation of reactive oxygen species, replicative stress, microtubule aberrations, and mitotic defects [45,46,47]. 

Irreversible and wholesale permeabilization of the mitochondrial outer membrane is a decisive event that culminates in intrinsic apoptosis [48]. Owing to the unique vesicular structure of the mitochondria, there is a gap between the inner and outer membranes, known as the intermembrane space, while the space enclosed by the inner membrane is called the mitochondrial matrix or inner chamber. Within the inner mitochondrial membrane, protons are pushed from the matrix into the gap between the two membranes, creating a proton gradient that accumulates a large number of unevenly distributed protons. Consequently, the outer chamber of the inner membrane is loaded with a high concentration of positive charges, whereas the inner chamber carries a significant number of negative charges. This generates mitochondrial membrane potential (MMP) [49], with the outer membrane carrying a positive charge and the inner membrane carrying a negative charge. Changes in MMP reflect the performance of the mitochondria, and their imbalance can lead to apoptosis and autophagy [50].

The Bcl-2 protein family primarily regulates mitochondria-associated apoptosis. Only pro-apoptotic proteins containing the BH3 structural domain (Bid, Bim, and Bad PUMA) activate the key pro-apoptotic proteins Bax and Bak, two Bcl-2 family proteins present only in mammals [51]. Bax is mainly free in the cytosol in normal cells and is activated to move to the outer mitochondrial membrane (OMM) [52]. Conversely, Bak is a transmembrane protein on the OMM, activated to oligomerize the dimer [53] and form a pore on the OMM, which allows the mitochondria to release AIF, cytochrome C (Cyt c), and a second mitochondrial activator of caspases (SMAC) [54].

The binding of released CytC with monomeric Apaf-1 induces oligomerization and the subsequently formed apoptosome mediates the conversion of pro-caspase-9 to activated caspase-9, which eventually activates effector caspase-3 and promotes apoptosis by binding to members of the inhibitor of apoptosis protein (IAP) family, such as X-linked inhibitor of apoptosis protein (XIAP) [55,56].

The principal anti-apoptotic proteins include B-cell lymphoma 2 (Bcl-2), B-cell lymphoma-extra-large (Bcl-xl), and induced myeloid leukemia cell differentiation protein 1 (Mcl-1) [57], which is exerted by blocking the activation of Bax and Bak through isolation of the structural domain of BH3; when their expression is decreased, the level of apoptosis increases [58,59]. Bcl-2 family protein downregulation induces cell apoptosis. In other words, the ratio of anti-apoptotic Bcl-2 proteins to pro-apoptotic Bcl-2 proteins determines apoptosis.

### 4.2. Extrinsic Death Receptor Pathway

Extrinsic apoptosis is triggered by the binding of extracellular ligand signals to death receptors (DRs) on the cell membrane [60]. Dependent receptors (e.g., deleted in colorectal cancer (DCC) and patched 1 (PTCH1)) are activated when their specific ligand levels fall below a specific threshold. Under physiological conditions, they exhibit anti-apoptotic effects. However, they trigger apoptosis when their ligands fall below a certain threshold [12]. Death receptors are members of the tumor necrosis factor (TNF) superfamily and include TNF receptor 1 (TNFR1), Fas (Apo-1 and CD95), TNF-related apoptosis-inducing ligand (TRAIL) receptors 1 and 2 (TRAIL-R1/2; death receptor 4/5 [DR4/5]), DR3, and DR6 [60].

Extrinsic apoptosis relies on the formation of DISC, and in the case of TNFR1 activation, tumor necrosis factor receptor type 1-associated DEATH domain (TRADD) is subsequently activated as an adapter protein, which in turn recruits Fas-associated death domain protein (FADD) [61]. All death receptors, except for TNF-R1 and DR3, directly activate FADD, an adaptor protein containing a C-terminal death effector domain (DD) and an N-terminal death effector domain (DED). One of the DEDs within procaspase-8 binds to the exposed DED of FADD associated with death receptors and assembles into DISC [62,63]. 

DED facilitates the oligomerization and recruitment of procaspase-8, leading to an increase in its localized concentration. Subsequently, procaspase-8 undergoes dimerization and autocatalytic cleavage, resulting in its conversion into the activated form. This activated caspase-8 serves as a crucial initiator in initiating downstream apoptotic signaling cascades, ultimately culminating in apoptosis [63].

In addition to regulating DISC function, cellular FLICE-like inhibitory protein (cFLIP) is structurally homologous to caspase-8 but devoid of proteolytic activity; cFLIPL contains two N-terminal DEDs and a C-terminal cysteinase-like structural domain. Since cFLIPL has a stronger ability to heterodimerize with procaspase-8 than itself, cFLIPL amplifies the effect of DISC-mediated apoptosis at physiological concentrations. cFLIPL, in excess, acts similarly to cFLIPL, blocking the DISC-mediated oligomerization of procaspase-8 and reducing the production of the activated apoptosis initiator, caspase-8 [62]. cFLIPS consists of only two DEDs without the cysteine-like structural domain. When cFLIPS is present, its heterodimerization with procaspase-8 blocks the homodimerization of procaspase-8 itself, thus exerting an inhibitory effect on apoptosis [64].

## 5. Hair Follicle Cycle and Cellular Changes

Simon et al. [65] indicated that the cells contained in the hair follicle, excluding differentiated hair follicle stem cells and miscellaneous cells such as immune cells, vascular cells, and erythrocytes, can be divided into five main categories: keratinocytes, melanocytes, neural crest cells, fibroblasts, and fibroblast-like cells. Hair formation originates from stem cell differentiation driven by ectodermal–mesodermal interactions. During this process, epithelial cells in the ectoderm receive signals from the dermis to thicken and form hair placodes. In response, mesenchymal cells produce dermal condensate [66]. After the dermis signals the epithelium again, ectodermally differentiated epithelial hair follicle stem cells (eHFSCs) proliferate downward and eventually wrap around the dermal condensate.

They remain in the upper bulge of the hair follicle [67]. The dermal condensate is wrapped in epithelial cells that differentiate into dermal papillae (DP). Melanocyte stem cells, differentiated from ectodermal-derived neural crest cells, remain in the bulge and replenish the bulb with melanocytes during the subsequent hair cycle [68]. Under epithelial–mesenchymal interactions, epithelial cells of ectodermal origin gradually differentiate to form the hair shaft, inner root sheath, and outer root sheath; mesoderm mesenchymal cells form the dermal papilla [69,70]; and both types of cells construct the basic structure of the hair follicle. 

Hair follicle stem cells (HFSCs) in the upper part of the hair follicle bulge are mainly used as backup energy sources for hair follicle regeneration. As the hair bulb continues to shrink and move upward until it encounters the HFSC, a second germ is formed, and the hair bulb moves back towards the derma [71]; however, the second germ originates from stem cells of the hair bulb rather than the rongeur stem cells and provides the main force for the transition from the catagen to the anagen phase [72,73,74]. Inward, the outer root sheath comprises primarily keratinized epithelial keratinocytes; its differentiation begins with epidermal hair follicle stem cells (eHFSCs) in the bulge region, supported by hair matrix cells in the hair bulb during follicle formation [75], serving as a protective layer for the hair shaft. The outer root sheath is followed by the inner root sheath, which is divided into three layers, the cuticle, Huxley’s layer, and Henle’s layer, followed by the hair shaft, where the hair cortex is the main body. The coloration of the hair is imparted by melanin granules produced by melanocytes in the hair bulb region. These granules fill the keratinocytes of the hair cortex [76].

The upper part of the hair follicle is a permanent structure, and changes are more pronounced in the lower part of the follicle during the hair cycle. The hair bulb contains hair matrix cells that undergo sustained proliferation and differentiation, ultimately culminating in the generation of inner and outer root sheaths, hair shafts, and assorted matrix cells [77].

Melanocytes derived from neural crest cells are present in the outer root sheath as a complement to epidermal melanocytes. The follicle bulge and secondary hair germ also contain melanocyte stem cells to replenish apoptotic melanocytes during the catagen phase [6,78]. DP is a mesenchymal cell of mesodermal origin responsible for signaling hair matrix cells to induce terminal differentiation. Hair follicles cannot undergo regeneration after removal of DP and return to the anagen phase after restoration of DP [79,80]. At the beginning of the anagen phase, β-catenin signaling produced by DP induces hair matrix cells to regulate the proliferation of keratinocytes at the hair bulb. Reducing the expression of β-catenin in DP using Cre recombinase resulted in dramatically shorter and sparser hair on mice and significantly decreased the proliferation rate of matrix cells at the hair bulb [81], demonstrating the necessity of DP for hair follicle maturation. Additionally, DP recruits and maintains melanocytes by converting dark-black eumelanin to yellow pheomelanin via the transmembrane protease corin [82]. The DS located in the outermost layer is a reservoir of DP cells, which contain dermal stem cells that self-renew to regenerate DS while contributing new cells to DP [83,84]; DS contraction is essential for follicle regression and DP niche relocation to stem cells [85].

Hair renovation is a cyclic progression; the hair cycle is divided into four phases: anagen (cell proliferation phase), catagen (apoptosis phase), telogen (resting phase), and exogen (hair shaft exfoliation). For humans, the cycle of hair follicles is approximately 2–8 years, and human hair follicles are asynchronous; therefore, according to the distribution of the hair follicle cycle, at any point in time, human hair is approximately 86% anagen follicles, 1% catagen, and 13% telogen [77,86]. In contrast, hair follicles in mice have a short cycle of approximately 30 days, in which the follicles are synchronized; after depilation of 2-week-old C57BL/6 mice using pine wax, the hair in the depilated area can be induced to enter the anagen phase at the same time [76].

## 6. Role of Apoptosis in Hair Cycle

During the hair cycle, apoptosis and proliferation occur in parallel, and cells are constantly replenished to complete this process. In normal physiological processes, apoptosis primarily occurs during the catagen phase. However, diseases primarily affect anagen hair follicles, leading to their targeted destruction. Figure 3 illustrates the factors involved in apoptosis during the hair cycle and the structural changes in the hair follicle cycle.

During the hair cycle, hair growth and renewal are regulated by several signals. For example, most cells in the hair follicle undergo apoptosis during the catagen phase, among which matrix cells in the hair bulb and stem cells in the bulge are more affected. Under normal conditions, matrix cells with high proliferative activity can proliferate through intercellular communication after the DP cells receive signals such as *Shh*, *BMP*, *Notch*, and *Wnt/β-Catenin* [87,88,89], which is the source of maintenance of the hair follicle during the anagen phase. In contrast, catagen onset may be attributed to the absence of growth factors or the reception of signals transmitted by death receptors [90,91]. Usually, apoptosis occurs as an indispensable step in maintaining the long-term renewal of the hair follicles, but some external influences, such as chemotherapeutic drugs, UV radiation, and viruses, can stimulate apoptosis of hair follicle cells, resulting in abnormal apoptosis of HFs [92] and, occasionally, permanent damage to the hair. For example, permanent hair loss caused by the use of the chemotherapeutic drugs leucovorin and doxycycline is attributed to the apoptosis of keratinocytes [93], while sequential administration of fluorouracil and docetaxel increases the possibility of permanent hair loss [94]. Depletion of HFSC is the primary cause of permanent chemotherapy-induced alopecia (PCIA), while leucovorin and cyclophosphamide (CYP) cause permanent hair loss through *PI3K/Akt* pathway-induced HFSC activation, in which the activation of *P53/P38* induces cell death, causing the depletion of HFSC and ultimately PCIA [95].

During the catagen phase, outer root sheath cells in the hair follicle highly express *TGF-β1/2* receptor and *Fas*, which activate extrinsic apoptosis. The lack of the anti-apoptotic protein Bcl-2 in the outer root sheath also affects the stability of the mitochondrial membrane potential [90]. *Gsdm3* is specifically expressed in the hair follicle; *Gsdm3* mutant mice showed delayed regression and reduced *Fas* receptor expression in the inner root sheath cells [96]. *L-cathepsin*-deficient and *hairless*-deficient mice exhibited slow growth and reduced apoptosis in the inner root sheath, demonstrating that *L-cathepsin*, the *hairless* gene, may be associated with inner root sheath apoptosis [97,98]. Ectodysplasin (Eda) and *Edar* receptor signals are expressed in the inner and outer root sheaths during regression, with a large reduction in XIAP expression, which suggests that Edar target signals are associated with the inhibition of apoptosis [99]. While the keratinocyte stroma-forming cells at the hair bulb were affected by TNF-α, TUNEL+ cells increased, along with a corresponding decrease in the ratio of Bcl-2/Bax compared with the early anagen. CIP/KIP family member proteins are locally expressed in matrix keratinocyte cells, controlling the cell cycle by balancing cyclin E and acting as an anti-apoptotic agent [100,101]. *P53* transcription factor is a double agonist of intrinsic and extrinsic apoptosis, and *P53*-deficient mice have fewer apoptotic cells in the catagen phase [101]. Specifically, dermal papilla and hair follicle stem cells antagonize apoptosis by overexpressing Bcl-2 while reducing the sustained expression of P53 to prevent apoptosis [102,103]. Typically, in the process of inhibiting hair loss, drugs exert their effects on the entire hair follicle. They downregulate several key targets including Fas, FasL, p53, Bax, active caspase-3, and procaspase-9. Concurrently, they facilitate the upregulation of Bcl-2 [104,105].

During the regression period, brain-derived neurotrophic factor (BDNF) activates TGF-β to induce a transformation between the growth and regression phases and promotes tumor regression by binding to TrkB. The p75 neurotrophin receptor (p75 NTR) binds to β-nerve growth factor (β-NGF) to regulate apoptosis-induced hair follicle degeneration [44].

## 7. Drug Treatments for Hair Loss: Targeting Apoptosis and Signaling Pathways

Alopecia is classified into either cicatricial or non-cicatricial. Cicatricial alopecia includes AGA, chemotherapy-induced alopecia areata, and telogen effluvium, while non-cicatricial alopecia includes lichen planopiloaris, frontal fibrosing alopecia, and lupus erythematosus [106]. Both classes are closely associated with apoptosis of hair follicle cells, most of which correlate with apoptosis of HFSCs. However, in some specific cases of hair loss, multiple factors are involved, such as genetics and the hair follicle microenvironment [107,108,109,110,111].

### 7.1. Androgenetic Alopecia

Androgenetic alopecia (AGA), the most common type of alopecia, is a heritable disease characterized by androgen dependence and a combination of factors in which hair follicles are progressively miniaturized by androgen, leading to hair reduction and thinning, which can occur in males and females [111,112]. Deng [113] has previously shown that the androgen receptor (AR) in DP cells mediated the paracrine secretion of *TGF-β* signaling and induced apoptosis of microvascular endothelial cells, thus making it difficult for hair follicles to enter the anagen phase. Xie et al. [114] showed that androgenetic alopecia-induced shrinkage or loss of the hair shaft caused mechanical compression, while the activation of *Piezo1* channels triggered Ca^2+^ inward flow, thereby increasing *TNF-α* sensitivity and inducing long-term apoptosis in HFSCs. 

*P53* expression is significantly increased in the hair follicles of the frontal baldness area in patients with AGA, whereas the expressions of the DNA repair marker proteins *APE1*, *PCNA*, and *PARP-1* were significantly reduced. This suggests that the DNA repair capacity during AGA is not sufficient to counteract the effects of apoptosis [115]. Conventional treatments for AGA include minoxidil and finasteride. Minoxidil restores hair growth mainly by promoting the microvascular concentration of hair follicles to increase the concentration of growth factors. Finasteride is an inhibitor of 5α-reductase, a key enzyme in the conversion of androgens in vivo, and reduces the conversion of androgenic testosterone to its active form, dihydrotestosterone [116]. Treatment with natural product extracts or monomers has also been used, such as proanthocyanidins, which inhibit hair epithelial cell apoptosis and thus promote hair growth [117]. Ginsenoside F2 reduces hair cells apoptosis by modulating *TGF-β2* in DHT-induced AGA mice and HaCaT cell models [118]. Baicalin and linoleic acid reduce hair cell apoptosis by regulating *IGF-1* [119,120], while forsythiaside-A, which inhibits hair cell apoptosis by decreasing the expression of caspases-9 and -3 and TGF-β2, delays the entry into the catagen phase [121].

### 7.2. Alopecia Areata

Alopecia areata (AA) is characterized by common non-scarring alopecia due to autoimmune disorders. Collapse of the immune privilege (IP) of the hair follicle bulb is an important driver of baldness, manifested by lymphocyte infiltration around the hair follicle during the anagen phase [122]. In immune therapy, drugs such as diphencyprone are used to reduce the aggregation of immune cells around the hair follicles [123]. By acting as allergens, diphencyprone induces a localized immune response in small areas of the scalp, away from the follicular region. Local subcutaneous injections of steroids are also used in treatment, along with emerging therapies like Janus kinase (JAK) inhibitors that directly inhibit downstream effectors of the immune response [124]. Additionally, the use of diphencyprone slightly increased the expression of apoptosis inhibitors p16 and survivin in AA patients [125], and Bcl-2 expression was significantly increased in recovered AA patients’ scalps [126].

However, there is also a strong genetic association with the development of AA, with significant alterations in the apoptosis/autophagy pathway [127]. In a case–control analysis, *FAS* and *FASLG* gene polymorphisms affected the risk of developing AA; that is, mutations in FAS and FASLG may be a cause of AA [128]. *FAS* and *FASL*-deficient mice are resistant to the development of AA, and AA mice showed more *FASL*-positive cells than control mice, possibly due to increased *FASL* gene expression during AA, which caused apoptosis of hair follicle cells [129].

### 7.3. Chemotherapy-Induced Alopecia

Chemotherapy-induced alopecia (CIA) is a common adverse effect in patients receiving chemotherapy and manifests as patchy or diffuse anagen alopecia that may progress to complete hair loss within 2–3 months [130]. There are two forms of damage attributed to CIA: One is the dystrophic anagen phase, in which hairs are stimulated by low doses of chemotherapeutic drugs, manifesting in an outgrowth of the hair shaft and damage to the structure of the hair follicle, which returns to the second cycle, during which hair loss is rare. The other is the dystrophic catagen phase, which refers to the destruction of the hair follicle structure after receiving high doses of chemotherapeutic drugs, characterized by the fragmentation of melanocytes, rapid and massive hair loss, and a rapid subsequent growth cycle within a short period of time [131].

A common clinical treatment option is scalp cooling to reduce drug damage to hair follicles by reducing the uptake and metabolism of local chemotherapeutic drugs. The active metabolite of CYP, 4-hydroperoxycyclophosphamide 4-HC, causes DNA damage and apoptosis, while *PPAR-γ* agonist NAGED pretreatment prevented 4-HC-induced apoptosis in human eHFSCs [132]. Some herbal extracts also have good therapeutic effects on CIA. Ginseng is commonly applied in the treatment of hair loss; for example, Korean red ginseng extract (KRGE) can prevent 4-HC-induced follicular anagen inhibition and premature catagen by reducing P53 and Bax/Bcl2 expression. KRGE also alleviates 4-HC-induced proliferation inhibition and apoptosis in matrix keratinocyte cells [133]. Monomeric compounds from Chinese herbal medicine also have good prospects for development, such as the decursin in Angelica, which can reduce the expression of *PI3K/AKT* signaling and *MAPK* signaling pathways in TNF-α-stimulated keratinocytes. Decursin has also been shown to reduce the expression of apoptosis-related factors and caspase family, while increasing the expression of epidermal growth factor, with certain therapeutic effects against CIA [134].

### 7.4. Primary Lymphocytic Cicatricial Alopecia

Lichen planopilaris (LPP) and frontal fibrosing alopecia (FFA) are primary lymphocytic cicatricial hair loss disorders. Similar to AA, hair follicles attacked by CD8+ cells due to the loss of IP at the bundle site undergo apoptosis and pathological epithelial-to-mesenchymal transition (EMT) of stem cells, and eventually loss of the regenerative capacity of the hair follicles [135]. HFSCs within the bulge are depleted, which is reflected in the tissue examination of the scalp of patients with LPP, where HFSCs showed an increase in apoptosis [136]. In contrast, FFA is more oriented toward pathological EMT and hair fibrosis [137].

Therefore, the most direct route to treat LPP and FFA is to inhibit IP collapse and reduce HFSC apoptosis with drugs such as tacrolimus that inhibits IP collapse in the bulb in vitro [138] and has undergone clinical trials [139]. In addition, agonists of the *PPAR-γ* pathway can partially reverse EMT at the bulb and are effective in isolated LPP hair follicles [140] and protect K15+ eHFSCs in the bulge region of LPP hair follicles from apoptotic injury, while reducing the number of CD8+ cells and MHCII+ cells among them [141].

Currently, various pharmaceutical interventions are being investigated to treat these diseases. Drugs that specifically target and regulate apoptosis can be classified into natural remedies and extracts, chemically synthesized, or biologically prepared (Table 1).

## 8. Conclusions

Studies on the relationship between hair loss and apoptosis have primarily focused on androgenic alopecia and CIA. The main factors influencing AGA include the AR protein and 5-α reductase, two proteins that are tightly linked to apoptosis. CIA has been extensively studied owing to its close association with apoptosis. Conversely, AA is largely associated with the loss of immune privilege and focuses on inflammation. Similar to AA, LPP and FFA emphasized IP collapse with increased EMT and loss of HFSCs. Both these factors increase cell apoptosis and warrant further investigations.

Apoptosis is stimulated by various intrinsic and extrinsic factors. In normal human metabolism, apoptosis is responsible for removing damaged or excess cells to maintain homeostasis; however, uncontrolled, excessive apoptosis is the cause of many diseases such as lupus erythematosus, an autoimmune disease, and hair loss during chemotherapy. The skin is the largest accessory organ of the human body; the hair follicle deserves attention since its growth and development are a microcosm of progenitor cell differentiation, and the stem cell pool contained therein has been studied in the field of skin grafting and hair transplantation. Notably, apoptosis is sometimes a manifestation, and the ultimate gene regulation process may be different; however, apoptosis occurs frequently, which justifies related research investigating the treatment of hair-related diseases. Therefore, the relationship between hair growth and apoptosis must be further investigated. 

When addressing the regulation of apoptosis in hair loss, the predominant emphasis has traditionally been placed solely on the hair follicle. Nevertheless, emerging research suggests that stem cells derived from the sebaceous gland additionally contribute significantly to hair growth. Furthermore, the PPAR pathway, known for its involvement in fat metabolism, exerts an impact on the hair follicle’s stem cells located in the hair bulge region. These diverse findings collectively indicate that exploring the surrounding appendages of the hair follicle could potentially offer novel strategies for addressing hair loss.

Gaining a deeper understanding of the relationship between apoptosis and hair loss is crucial for revealing the mechanisms underlying hair loss, discovering new therapeutic approaches, and aiding in the development of preventative strategies. There are limitations in the existing research regarding the association between alopecia and apoptosis. Therefore, subsequent studies should elucidate the mechanisms underlying apoptosis in various forms of alopecia, identify novel targets for modulating apoptosis, and develop treatment modalities targeting apoptosis.

## Figures and Tables

**Figure 1 biomolecules-14-00020-f001:**
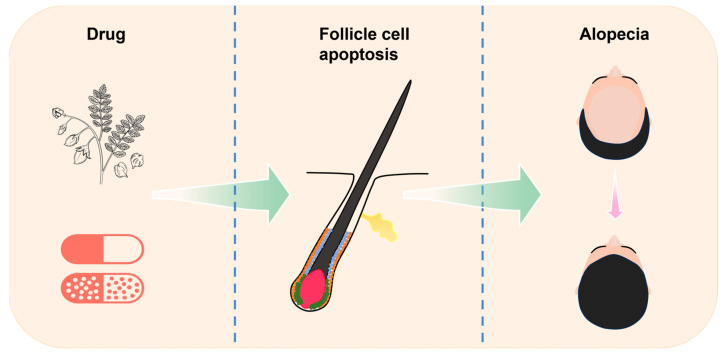
Relationship between ingredients, apoptosis, and alopecia. This article introduces the process of cell apoptosis and the cycle of hair follicle, with a focus on elucidating the apoptosis that occurs in the hair follicle. By summarizing various natural herb extracts and chemical drugs that can regulate apoptosis and their applications in hair loss diseases, the article provides some new insights for subsequent research on hair loss prevention and treatment.

**Figure 2 biomolecules-14-00020-f002:**
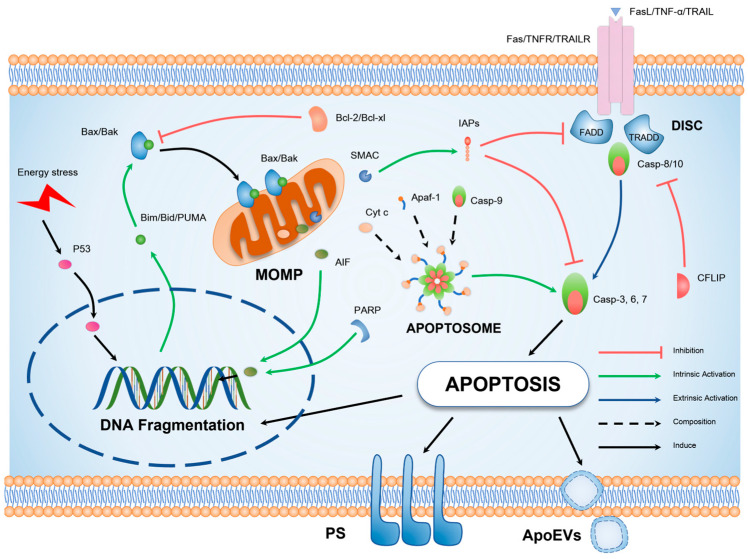
Schematic illustration of the molecular mechanism underlying apoptosis. Cellular apoptosis can be elicited by exogenous factors such as tumor necrosis factor-alpha (TNF-α) signaling or endogenous DNA damage. The extrinsic apoptotic pathway is primarily characterized by oligomerization of death receptors and formation of DISC, in which the key pro-apoptotic protein is the initiator caspase-8. The intrinsic apoptotic pathway, on the other hand, is distinguished by mitochondrial outer MOMP, representing the translocation of BCL-2 (B-cell lymphoma 2), homology domain 3 (BH3), interacting domain death agonist (BAX), and BCL-2 antagonist/killer (BAK) proteins to the outer mitochondrial membrane. This results in the release of pro-apoptotic factors and the generation of apoptotic bodies, culminating in the activation of caspase-3. Both extrinsic and intrinsic pathways ultimately culminate in the hallmark characteristics of apoptotic cells, including DNA fragmentation, formation of apoptotic bodies, and externalization of phosphatidylserine.

**Figure 3 biomolecules-14-00020-f003:**
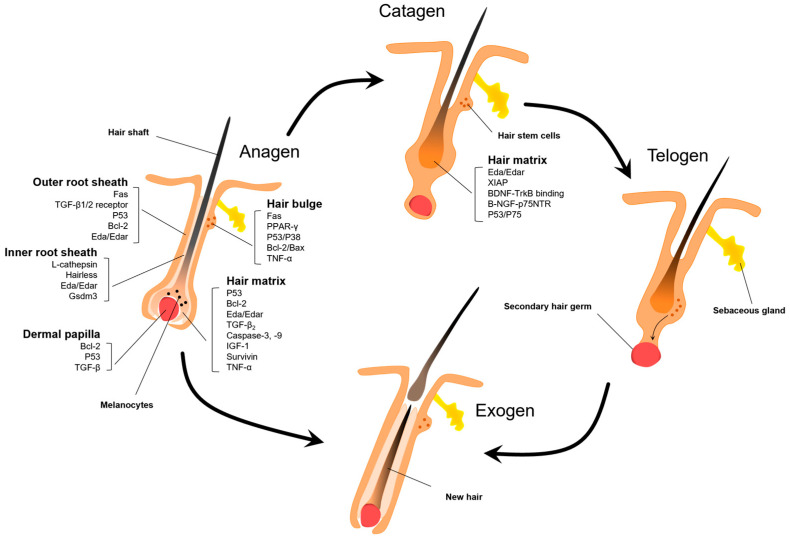
Apoptotic factors during the hair follicle cycle. Apoptosis plays a critical regulatory role in the anagen and catagen phases of the hair follicle cycle. Due to their rapid cellular proliferation, anagen hair follicles are particularly susceptible to apoptosis upon stimulation with exogenous factors. The key apoptotic cell populations encompass keratinocytes of the outer root sheath and hair matrix, bulge hair follicle stem cells, as well as dermal papilla cells of the hair bulb, involving apoptotic proteins such as *P53*, *Fas*, *Bcl-2*, and transforming growth factor beta (TGF-β). Spontaneous apoptosis occurs in various segments of hair follicles during the catagen phase. Although current therapeutic approaches targeting follicle cells apoptosis have focused on inhibiting hair bulge and matrix apoptosis, interventions at this juncture may be limited.

**Table 1 biomolecules-14-00020-t001:** Pharmaceutical interventions and regulatory factors.

Classification	Ingredient	Study Model	Effect	Mechanism	Reference
AA	Diphencyprone	Human skin	Increased microvessels	↑P16, ↑survivin	[125]
AA	Diphenylcyclopropenone	C57BL/6 mice, Human skin	Autoreactive T cell activation induced cell death	↑FasL	[142]
AA	Diphencyprone	Human skin	Hair growth improved	↑Bcl-2	[126]
AA	Tofacitinib citrate delivered phospholipid calcium carbonate hybrid nanoparticles	C57BL/6 mice	Apoptosis restrained in CYP-induced follicles cells, AA relieved	-	[143]
AGA	Caizhixuan hair tonic	C57BL/6 mice	Hair regrowth improved	↓Caspase-3, Bax; ↑Bcl-2	[144]
AGA	Policosanol	KM mice, Human follicle dermal papilla cells (HFDPCS)	HFDPCS apoptosis increased	↓TGF-β2, cleaved caspase-9, cleaved caspase-3, Bax; ↑Bcl-2	[145]
AGA	Ginsenoside F2	HFDPCS, HaCaT cells, C57BL/6 mice	HFDPCS and HaCaT cell proliferation increased, hair cell apoptosis and premature entry to catagen suppressed	↓TGF-β2, cleaved caspase-3, Bax, caspase-12; ↑Bcl-2	[118]
AGA	Forsythiaside-A	HFDPCS, HaCaT cells, C57BL/6 mice	Mouse hair density and thickness increased, HFDPCS and HaCaT cells apoptosis suppressed	↓TGF-β2, caspase-9, caspase-3, Bax; ↑Bcl-2	[121]
AGA	*Stauntonia hexaphylla* Extract	C57BL/6 mice, HFDPCS	Inhibited 5α-reductase and AR resulting in reduced apoptosis and induced cell proliferation in HFDPCS	↓Bax/Bcl-2; ↑PARP-1	[146]
AGA	Finasteride-loaded microspheres	C57BL/6 mice	Finasteride-loaded microspheres for subcutaneous use significantly reduced testosterone-induced alopecia	↓TGF-β2, caspase-3	[147]
AGA	*Acanthus ebracteatus Vahl*. extract and verbascoside	HFDPCS, RAW 264.7 cells	Inhibited the release of pro-inflammatory cytokines from RAW 264.7 cells and HFDPCS prevented cell apoptosis induced by testosterone	↓TNF-α	[148]
AGA	Phospholipid–polymer hybrid nanoparticle-mediated transfollicular delivery of quercetin	SD rats	Hair regrowth potential improved, HFs cell apoptosis inhibited	-	[149]
AGA	Triton-modified polyethyleneimine conjugates assembled with growth arrest-specific protein 6	C57BL/6 mice	Transfected *Gas6* prolongs the anagen status, inhibited hair follicle cell apoptosis	↑Bcl-2	[150]
AGA	VEGF	HFSCS	Reversed the 5α-DHT-induced apoptosis of HFSCS	↑Bcl-2/Bax; ↓caspase-3	[151]
AGA	Sulforaphane, glucosinlates, *Brassica oleracea* L. var. *italica* Planch extract	HaCaT cells, DPC	Enhanced DPC and HaCaT cells viability	↓Bax	[152]
CIA	Decursin	C57BL/6 mice, HaCaT cells	Recovered dystrophic hair follicles, hair regeneration restoration	↓Caspases -3, -7, and -8	[134]
CIA	N-acetyl-GED-0507-34-Levo	Human hair follicle	Reduced eHFSC DNA damage and EMT	↓P53	[132]
CIA	α-lipoic acid derivative	C57BL/6 mice	Decreased vascular endothelial cell apoptosis, enhanced vascular permeability	↑IGF-1	[153]
CIA	Palbociclib	Human hair follicle	Suppressed hair matrix keratinocyte apoptosis induced by stem cell injury	↓Caspase-3	[154]
CIA	YH0618	C57BL/6 mice	Inhibited alopecia	↓Bax/Bcl-2	[155]
CIA	Human placenta	C57BL/6 mice	Increased *Ki67*-positive cells in hair follicles	↑Bcl-2/Bax; ↓P53, Cyt c, caspases-3 and -9	[156]
CIA	Shh protein	C57BL/6 mice, human skin	Shh protein partially rescued hair loss	↓caspase-3	[157]

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
