# Peer review of "Controlling Hair Loss by Regulating Apoptosis in Hair Follicles: A Comprehensive Overview"

_biomolecules, 2023, doi:10.3390/biom14010020_

Round 1

Reviewer 1 Report

Comments and Suggestions for Authors

Dear authors,

The authors have presented a comprehensive exploration of apoptosis mechanisms and their relevance to both the hair follicle and hair loss. Additionally, they have summarized recent studies to elucidate drug treatment options for addressing hair loss. This information is pivotal for comprehending the intricacies of hair loss mechanisms and for the development of therapeutic strategies and medications. I recommend that the following aspects be taken into consideration prior to publication.  

P3, L134-135:

Please add citation regarding this sentence.

P4, Figure 1:

The authors are requested to incorporate explanatory sentences for Figure 1 within the main body of the text. Please more clarify relationships between Figure 1 and main body of the text.

P6, L268:

I am unclear about the knockout of the catenin allele. If it is asserted that beta-catenin is crucial for hair follicle maturation, I suggest splitting this sentence into two for improved clarity. 

P10, Table 1:

Please take care about line break (e.g.: classificatio-n).

Finally, as a reader, I found it challenging to comprehend the content solely through text descriptions. If feasible, I suggest incorporating figures to visually elucidate the main body of the text for enhanced understanding.

Reviewer 2 Report

Comments and Suggestions for Authors

The chart on pages 11-12 needs lines or spaces to separate each subject as the text from one row flows into the next row.  

This is a complex and intricate manuscript. If I take the science presented here for face value, the manuscript seems well-organized and well-put together.  The Chart on pages 11-2 has some clinical relevance to my practice. 

Author Response

Response to comment: The chart on pages 11-12 needs lines or spaces to separate each subject as the text from one row flows into the next row.

Response: Based on your reminder, I have increased the spacing between each item, which avoids crowding between the texts and improves the readability of the chart.

Special thanks to you for your good comments.

Reviewer 3 Report

Comments and Suggestions for Authors

Dear Authors,

Thank you very much for the comprehensive review on apoptosis and the influence of the biological process on hair loss condition. Despite positive overall impression, there are several minor corrections should be done prior to the publication possibility, namely:

lines 11-12 the sentence '..., significantly affects hair' should be reformulated since the impression that apoptosis mainly related hair, whereas the apoptosis is the key process of living cells.

lines 17-18 the sentence 'Hair follicles undergo cyclic periods of growth, 17 regression, and rest.' should be also reformulated since term 'rest' is not related cell cycle, and the terms 'interphase, dormacy' and so on is more appropriate in the case.

figure 2. please correct misprint 'Telegen' to 'Telogen'

Also, a there is a term 'mitoptosis' that designate used in the manuscript term  'intrinsic apoptosis' at least by part; however the authors are free to use this term up to their choice.

Additionally, there are papers on strong interrelation between hair follicle microenvironment and hair loss condition; unfortunately this review not encompasses this aspect of the issue. 

Comments on the Quality of English Language

The are just a little correction needs to be done, mainly related using uncorrect terminology like:

lines 11-12 the sentence '..., significantly affects hair' should be reformulated since the impression that apoptosis mainly related hair, whereas the apoptosis is the key process of living cells.

lines 17-18 the sentence 'Hair follicles undergo cyclic periods of growth, 17 regression, and rest.' should be also reformulated since term 'rest' is not related cell cycle, and the terms 'interphase, dormacy' and so on is more appropriate in the case.
